# A Two-Step Framework to Recognize Emotion Using the Combinations of Adjacent Frequency Bands of EEG

Zhipeng Zhang and Liyi Zhang *

School of Information and Management, Wuhan University, No. 16, Luojiashan Road, Wuchang District, Wuhan 430072, China
* Correspondence: lyzhang@whu.edu.cn

**Abstract:** Electroencephalography (EEG)-based emotion recognition technologies can effectively help robots to perceive human behavior, which have attracted extensive attention in human–machine interaction (HMI). Due to the complexity of EEG data, current researchers tend to extract different types of hand-crafted features and connect all frequency bands for further study. However, this may result in the loss of some discriminative information of frequency band combinations and make the classification models unable to obtain the best results. In order to recognize emotions accurately, this paper designs a novel EEG-based emotion recognition framework using complementary information of frequency bands. First, after the features of the preprocessed EEG data are extracted, the combinations of all the adjacent frequency bands in different scales are obtained through permutation and reorganization. Subsequently, the improved classification method, homogeneous-collaboration-representation-based classification, is used to obtain the classification results of each combination. Finally, the circular multi-grained ensemble learning method is put forward to re-exact the characteristics of each result and merge the machine learning methods and simple majority voting for the decision fusion. In the experiment, the classification accuracies of our framework in arousal and valence on the DEAP database are 95.09% and 94.38% respectively, and that in the four classification problems on the SEED IV database is 96.37%.

**Keywords:** electroencephalogram; emotion recognition; homogeneous collaboration representation; circular multi-grained scanning; ensemble learning

## 1. Introduction

Emotion is human beings' subjective consciousness that can reflect their current physiological and psychological state and affect their cognitive process, communication, and decision-making ability in daily life [1]. Many studies indicate that emotion recognition can improve the communication quality between humans and intelligent devices, so the automatic recognition of emotional states has become indispensable [2,3]. In general, emotion recognition methods rely on physiological data such as blood pressure, electrocardiogram (ECG), and functional magnetic resonance imaging (FMRI), as well as non-physiological data such as eye movements, expressions, and speech [4,5]. Relatively, methods based on physiological data typically produce better results because they are less susceptible to subjective will.

As a physiological signal that records electrical changes in brain activity, EEG signals have received a lot of attention in neuroscience, psychology, and clinical medicine due to their ability to capture and reflect emotional states in real time, which enables relevant researchers to obtain convincing and unbiased results. Consequently, EEG signals are widely used in engineering, education, and medical research [6–8].

Although promising results have been obtained with regard to EEG-based emotion recognition methods, it remains a challenge to integrate useful EEG information to improve machine learning prediction [9]. The interesting frequency range of EEG signals can be

divided into five frequency bands based on the rhythmic characteristics. As the information interaction of brain waves is a cross-frequency coupling process among the frequency bands [10,11], it is reasonable to take this interaction effect into account when establishing an EEG-based emotion recognition model. Additionally, from the perspective of statistical parametric maps (SPMs), the band energy of EEG signals has a certain correlation, and the frequency band beta is especially correlated with alpha [12]. This can prove that there is an interaction between different EEG frequency bands.

To utilize the interacted information from different frequency bands, the emotion recognition framework that uses the combinations of all the adjacent frequency bands is designed to mine the complementary information as much as possible. The entire process is shown in Figure 1. In the data preparation process, the raw EEG signals are preprocessed for power spectrum density (PSD) extraction. After that, in the first step of the framework, the combinations of the adjacent frequency bands are considered as a subset, each subset is divided into a training subset and testing subset, respectively, and each training subset and the corresponding testing subset are both tested by the homogeneous-collaboration-representation-based classification (HCRC) method. In the second step, the testing results of all the training subsets and testing subsets are spliced together, respectively, to form a testing decision set and a training decision set. Then, the circular multi-grained ensemble learning (CGEL) method is used to complete the decision fusion and obtain the final classification result of the testing decision set.

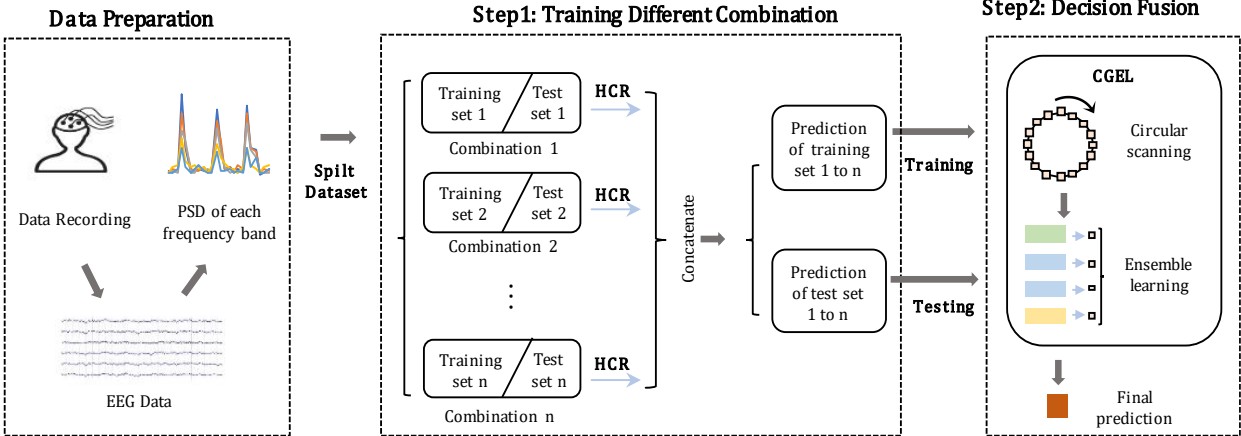

**Figure 1.** The flow chart of HCRC-CGEL framework.

The main contributions of our work include:

(a) The CRC_RLS method is optimized to retain the same dimension of the representation coefficient in each category;
(b) The CGEL decision fusion method is designed to improve the prediction accuracy;
(c) An EEG-based classification framework HCRC-CGEL is constructed to utilize the complementary information from different frequency bands;
(d) The experiments on two databases demonstrate the performance of the framework.

The remaining chapters of this paper are arranged as described below. We review the different types of EEG features, current EEG-based emotion recognition methods, and decision fusion methods in Section 2. We present the principle of HCRC-CGEL and the related concepts, including PSD, the HCRC method, and the architecture of the decision fusion method CGEL in Section 3. We present the used DEAP and SEED IV databases, the preprocessing processes, and the experimental results in Section 4. The conclusions and further work are demonstrated in Section 5.

## 2. Related Work

On the basis of EEG signals, the emotion recognition technologies mostly concentrate on extracting discriminative features and establishing effective emotion recognition models. The commonly used features have mainly been based on the Fourier transform (FT), wavelet transform (WT), statistics, and entropy [13–15]. These features have widely been used in current studies, and each has its own set of benefits.

Driven by data, Sun et al. [16] proposed a feature extraction method in which the EEG signals were encoded by an echo state network (ESN) and the features were extracted by the recurrent autoencoder, and this method was more effective than the current SOTA method. To save time and increase efficiency, Zhuang et al. [17] transformed the EEG signals into intrinsic mode functions (IMFs), and the multidimensional information of IMF, which has 8 channels, can be applied for emotion classification. To use the specificity of EEG channels, Gupta et al. [18] switched the EEG signals into different sub-bands by the flexible analytic WT so that the information potential could be applied for feature extraction by those sub-bands. To overcome the disadvantages of manual features, Hu et al. [19] proposed a ScalingNet that could dynamically generate many convolution kernels to make a spectral map from the original EEG signal for emotion recognition.

The development of a high-performance classifier is another important stage in the EEG-based emotion categorization model. Based on the convolutional neural networks (CNNs), the complex neural network can be designed to produce inspiring emotion recognition results [20–22]. For example, using an extended CNN model combined with spectrum theory, a graph CNN can learn structural information and different features at the same time [23]. The dynamic graph CNN, which is different from the traditional graph CNN, can take full advantage of the different channel information of EEG data through training the neural network and extracting more discriminative EEG features [24]. The long short-term memory (LSTM), which can avoid gradient disappearance in the algorithm backpropagation process, is applicable for dealing with time-related series problems [25]. The Bi-directional LSTM, which combines the forward and backward data of input on the basis of LSTM, can capture different characteristics through the embedded loop structure and acquire better classification results [26].

Despite the clear benefits of neural-network-based techniques, there are some drawbacks as well. The neural-network-based methods require a lot of training data, and the results largely rely on the adjustment of hyperparameters. However, due to the limitations of equipment, manpower, and other reasons, the EEG dataset with a large sample size is difficult to obtain. To solve this problem, the sparse representation-based classification (SRC) method uses the representation distance of the training set to minimize the regularized residual and determines the classification results by the category that could produce the minimum regularized residual. On the basis of SCR, the collaboration-representation-based classification with regularized least squares (CRC_RLS) method considers the difference in regularized residuals both in the target category and other categories, which has more stable results in pattern classification problems [27].

In decision fusion, the results of each classifier are connected independently to obtain the final result through some rules [28]. To preserve the maximal uniformity of decisions, the Dempster–Shafer (DS) method takes all the pieces of available decisions into account to combine the multimodal results [29]. Through the principle of minimum loss of the training set, adaptive weight learning integrates the decisions through assigning different weights to the results of each classifier [30]. The ensemble model, gcForest [31], which uses multi-grained scanning to re-extract features and build an adaptive cascade forest for representational learning, can automatically adjust the training process in the cascade forest layers and is insensitive to the setting of hyperparameters [32]. However, the edge data under the scanning rule would be ignored. Inspired by this problem, this paper designed an ensemble learning method that is more suitable for decision fusion.

## 3. Methods

### 3.1. Combinations of All the Adjacent Frequency Bands

The different frequency bands in EEG signals can reflect the specific brain consciousness of humans [33,34], and the performances of the EEG-based emotion recognition methods are closely related to the choice of the selected frequency bands [35]. Taking face recognition task as an example, Shen et al. [30] explained that faces can show obvious features and structures in different parts, and different parts provide complementary information to each other; reasoning by analogy, they considered that the combinations of all the adjacent frequency bands have different emotional characteristics and complement each other.

In view of the fact that different frequency bands contain identification and complementary information, we assume that the combinations of all the adjacent frequency bands have different emotional characteristics and complement each other. Based on this assumption, we arrange the five frequency bands of EEG signals in order, and to the adjacent frequency bands, there are 15 combinations in total with the combined scale ranging from one to five. For example, when we consider the situation of scale two, there are four combinations, such as delta with theta, theta with alpha, alpha with beta, and beta with gamma frequency bands.

### 3.2. The HCRC Method

CRC_RLS [27] is an unsupervised classification method that searches a representation coefficient vector to combine the training set with the shortest representation distance and determines the classification results by the categories with the minimal regularized residual. As the data of different emotions are correlated, the regularized residuals of each category are relatively small. If the sample numbers in different categories are uneven, the dimension of the representation coefficient will be unequal when used to generate the regularized residual of each category, and the classification result will be influenced to some extent. To achieve better classification results, this paper proposes the HCRC method to randomly select (and reject simultaneously) some samples to keep the sample number of each category constant.

Specifically, for a dataset of sample size $N$, consider a training set $X_{train}$ of a stimulus that has $n$ categories. The subset of the $i$-th category is noted as $X_i = [x_{i1}, x_{i2}, \dots, x_{in_i}]^T \in R^{n_i \times m}$, where $x_{ij} \in R^{m \times 1}$ represents the $j$-th data of the $i$-th category with $m$ elements, $j = 1, 2, \dots, n_i$ and $i = 1, 2, \dots, n$; then, the training set $X_{train}$ can be expressed as $X_{train} = [X_1^T, X_2^T, \dots, X_n^T]^T \in R^{(\sum n_i) \times m}$. As shown in Figure 2, the sample number $l$ of the category with the minimum sample size in $X_{train}$ is selected as the extraction number. For category $X_i$, the simple random sampling method is used to sample $l$ samples from it to compose the sampled data $\widetilde{X}_i$, and the sampled data from each category are combined together to compose the extracted training set $\widetilde{X} = [\widetilde{X}_1^T, \widetilde{X}_2^T, \dots, \widetilde{X}_n^T]^T \in R^{nl \times m}$.

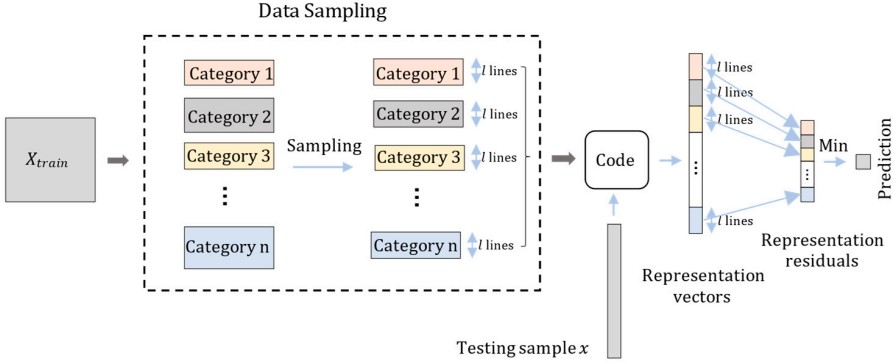

**Figure 2.** The flow chat of HCRC.



For a testing sample $x \in R^{m \times 1}$, the representation distance from it to $\overset{\sim}{X}$ can be express as

$$
\begin{aligned}
d &= \parallel x - \overset{\sim}{X}^T \rho \parallel_2^2 + \lambda \parallel \rho \parallel_2^2 \\
&= \parallel x - \sum_{i=1}^n \overset{\sim}{X_i}^T \rho_i \parallel_2^2 + \lambda \sum_{i=1}^n \parallel \rho_i \parallel_2^2
\end{aligned}
\tag{1}
$$

where $\rho = [\rho_1^T, \rho_2^T, \ldots, \rho_n^T]^T \in R^{nl \times 1}$ are the representation vectors of $\overset{\sim}{X}$, $\rho_i \in R^{l \times 1}$ is the representation coefficient of $\overset{\sim}{X_i}$, and $\lambda$ is a regularization parameter. In Formula (1), the minimum value of the representation distance $d$ can be calculated by the least squares estimation of $\rho$, which can be expressed as

$$
\begin{aligned}
\hat{\rho} &= \underset{\rho}{\mathrm{argmax}} \left\{ \parallel x - \overset{\sim}{X}^T \rho \parallel_2^2 + \lambda \parallel \rho \parallel_2^2 \right\} \\
&= \left( \overset{\sim}{X} \overset{\sim}{X}^T + \lambda I \right)^{-1} \overset{\sim}{X} x
\end{aligned}
\tag{2}
$$

where $\hat{\rho} \in R^{nl \times 1}$ can be written as $\hat{\rho} = [\hat{\rho}_1^T, \hat{\rho}_2^T, \ldots, \hat{\rho}_n^T]^T$, where $\hat{\rho}_i \in R^{l \times 1}$ is the estimated representation coefficient of $\overset{\sim}{X_i}$. Thus, the regularized residual of category $\overset{\sim}{X_i}$ $(i = 1, 2, \ldots, n)$ would be expressed as

$$
e_i = \parallel x - \overset{\sim}{X_i}^T \hat{\rho}_i \parallel_2 / \parallel \hat{\rho}_i \parallel_2
\tag{3}
$$

Each category has a regularized residual $e_i$, and the category with the minimum regularized residual is the classification result of the HCRC method. For each sampled $\overset{\sim}{X_i}$, under certain sparsity constraints, $\overset{\sim}{X_i}$ only needs a small number of samples to represent $x$, which proves that the imbalance of samples has little effect on the classification results. Therefore, it is suitable to process the training sets of different categories into the same numbers of samples, and this will make the dimension of $\hat{\rho}_i$ in each type of training set the same. In addition, the differences between the regularized residuals of each category are small, so the units of data magnitude may slightly influence the classification results. In order to eliminate that impact, we normalize the data before putting it into the HCRC method.

In the first step of our framework, for an EEG dataset, all the subsets of 15 adjacent combinations are divided into the training subset and testing subset consistently and respectively. Each training subset and testing subset has the corresponding prediction result using the HCLC method, and the prediction result is noted as a column vector. For the next step, the training decision set and testing decision set are made up by, respectively, concatenating the results of all the training subsets and testing subsets by column for decision fusion.

### 3.3. The CGEL Method

The decision fusion model CGEL is inspired by gcForest [31]. gcForest is designed to re-extract features and then achieve classification through ensemble learning of random forests. As the weight of the input data under the scanning rule of gcForest is uneven, CGEL uses a circular multi-grained scanning method to generate majority voting of the input samples. This gives equal weight to each element of the input sample and is more suitable for decision fusion. In order to learn the structure of the scanned samples, CGEL uses the ensemble learning of random forest (RF) [36], complete random forest (CRF) [37], decision tree (DT) [38], and simple majority voting (SMV) layer by layer to learn the most appropriate method for each cascade layer and the best number of the cascade layers.

The CGEL method can be divided into two steps: circular multi-grained scanning and ensemble learning. Circular multi-grained scanning is designed to generate majority voting features. In this section, the training decision set and testing decision set are noted as $D_{train} \in R^{(\sum n_i) \times 15}$ and $D_{test} \in R^{(N - \sum n_i) \times 15}$, respectively. The inputs of the circular multi-grained scanning are the samples of the decision sets.

As shown in Figure 3, for a sample $t$ from the decision sets, considering that each element of $t$ should have the same weight in the scanning process, $t$ is spliced back and forth to form a closed loop $t^*$. The closed loop $t^*$ is scanned by the sliding window of size $w_i \in R$ ($w_i$ is smaller than the number of combinations) and stride $s_i \in R(s_i = 1, 2, 3)$. In one scanning process, a $w_i$-dimensional scanned vector is produced by sliding the window for one stride, and the scanning process stops when the head of the sliding window slides a full circle in $t^*$. One scanning process produces one scanned set $W_i$ with several scanned vectors. To each scanned vector, there is a simple majority vote of its element; all the voted elements are spliced together to generate the scanned features $l_{w_i,s_i}$. By setting up three different sizes of sliding windows $w_i$ and strides $s_i$, all the generated scanned features in the three scanning process can be stitched together to compose the final concatenated vector $l^*_{w,s} = (l_{w_1,s_1}, l_{w_2,s_2}, l_{w_3,s_3}) \in R^{1 \times h}$, where $h$ is the dimension of the concatenated vector.

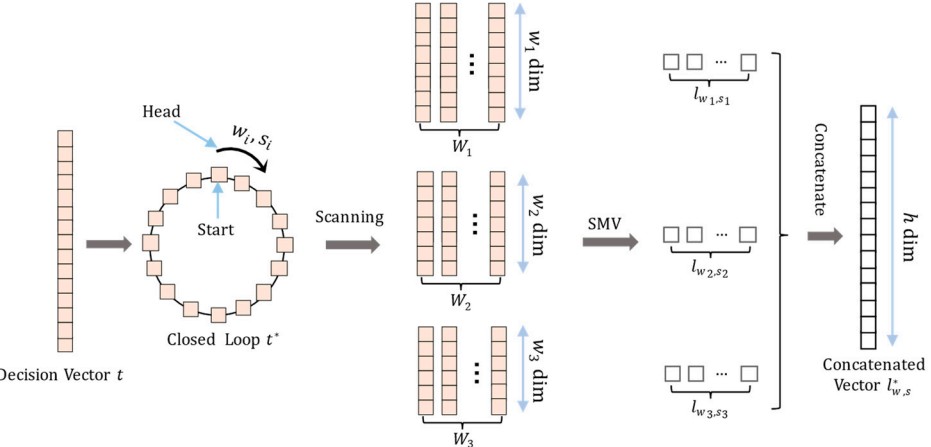

**Figure 3.** The flow chat of circular multi-grained scanning.

For example, to a decision vector $V_i = [0, 0, 1, 1, 3, 1, 0, 1, 2, 1, 1, 0, 0, 0, 0]$ with fifteen elements, each element is a classified result of a combination. If we use a window ($w_i = 12$) to scan $V_i$ with a stride ($s_i = 2$), the scanned set is

$$W_i = \{[0, 0, 1, 1, 3, 1, 0, 1, 2, 1, 1, 0], [1, 1, 3, 1, 0, 1, 2, 1, 1, 0, 0, 0], [3, 1, 0, 1, 2, 1, 1, 0, 0, 0, 0, 0],$$
$$[0, 1, 2, 1, 1, 0, 0, 0, 0, 0, 0, 1], [2, 1, 1, 0, 0, 0, 0, 0, 0, 1, 1, 3], [1, 0, 0, 0, 0, 0, 0, 1, 1, 3, 1, 0],$$
$$[0, 0, 0, 0, 0, 1, 1, 3, 1, 0, 1, 2], [0, 0, 0, 1, 1, 3, 1, 0, 1, 2, 1, 1]\}$$

where the elements of $W_i$ are scanned vectors, and all the simple majority vote results of the elements of scanned vectors are $\{1, 1, 0, 0, 0, 0, 0, 1\}$. Then, the scanned feature of one scanning process is $l_{w_i,s_i} = [1,1,0,0,0,0,0,1]$.

Inspired by the strategy of the level-by-level step of gcForest, ensemble learning adopts a cascade structure to acquire and transmit feature information through the procedure of cascade layers. Each layer is an ensemble of RF, CRF, DT, and SMV.

The training process of ensemble learning is shown in Figure 4. All the samples of the $D_{train}$ and $D_{test}$ are, respectively, scanned to generate the corresponding concatenated vectors to compose the training feature set $L_{train} \in R^{(\sum n_i) \times h}$ and testing feature set $L_{test} \in R^{(N - \sum n_i) \times h}$ before training. For $L_{train}$, four prediction results are generated by RF, CRF, DT, and SMV through self-training of $L_{train}$, and from the four results, the best result $p^0 \in R^{(\sum n_i) \times 1}$ with the highest prediction accuracy $a_0$ is chosen as the enhanced feature. $p^0$ is concatenated with $L_{train}$ to generate an enhanced training feature set $L^1_{train} = [L_{train}, p^0] \in R^{(\sum n_i) \times (h+1)}$ as the second layer input. There are also four predicted results in the second layer. If the best result $p^1 \in R^{(\sum n_i) \times 1}$ from the four results of the second layer with highest prediction accuracy $a_1$ is better than $a_0$, $p^1$ in this layer will be concatenated with $L^1_{train}$ to generate a new enhanced training feature set $L^2_{train} = [L^1_{train}, p^1] \in R^{(\sum n_i) \times (h+2)}$ as the next

layer input. In addition, the process continues until the highest prediction accuracy $a_i$ does not increase any longer or the number of iterations reaches a threshold.

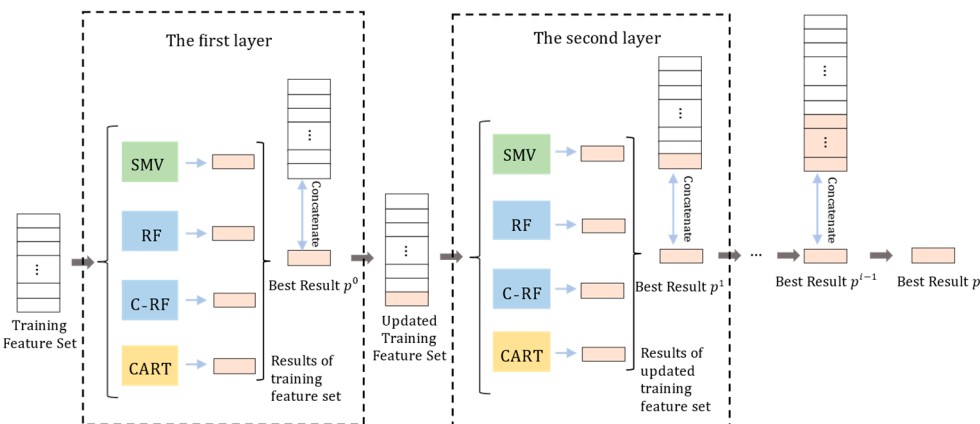

**Figure 4.** The training process of ensemble learning (each rectangle in the training feature set represents a concatenated vector).

Then, the number of layers stops increasing and the training stops. The best result in the last layer is the training result, and the method from RF, CRF, DT, and SMV to generate the best result $p^i$ in each layer is noted and used in the testing process. The final results of $L_{test}$ is computed through the trained number of layers and the corresponding noted method from RF, CRF, DT, and SMV to obtain the best results in each layer.

## 4. Materials and Experiments

### 4.1. Database Introduction and Preprocessing

To assess the effect of the emotion recognition framework, we conduct the experiments on two widely used public EEG databases, and Table 1 shows the particulars of the original and preprocessed data in both databases.

**Table 1.** The details of the pre-processed database. (Arousal and valence are represented by A and V, respectively; happy, sad, fear, and neutral are represented by H, S, F and N, respectively.)

| Database | Data Content | Data Shape | Data Description |
|----------|--------------|------------|------------------|
| DEAP | Raw data | $40 \times 32 \times (128 \times 63)$ | Video $\times$ channel $\times$ (sample rate $\times$ time) |
| | Preprocessed data | $W_1 \times 128$ | sample number $\times$ (channel $\times$ band) |
| | Preprocessed labels | $W_1 \times 2$ | sample number $\times$ (A, V) |
| | Number of dataset | 32 | trial |
| SEED IV | Raw data | $15 \times 62 \times (200 \times n_i)$ | video $\times$ channel $\times$ (sample rate $\times$ time) |
| | Preprocessed data | $W_2 \times 310$ | sample number $\times$ (channel $\times$ band) |
| | Preprocessed labels | $W_1 \times 4$ | sample number $\times$ (H, S, F and N) |
| | Number of dataset | 45 | trial |

The DEAP [39] database contains the EEG signals with 32 channels and a 512 Hz sampling rate from 32 subjects (50% female, mean age 26.9 years) watching 40 videos with a length of 63 s (a one-minute music video plus a three-second baseline). Each subject has to perform a trial in which they have to self-assess their arousal, valence, liking, and dominance in each music video, and the self-assessment score is the label of the current recorded EEG signal.

In our experiment, the preprocessing processes of the DEAP database are conducted by downsampling the EEG signals to 128 Hz, eliminating the artifacts, and filtering the signal into different frequency bands. Then, the PSD of each frequency band is extracted with a window size of three seconds and no overlap. Additionally, in the initial moment of each music video, there is a three-second baseline signal. Thus, each processed segment

has twenty-one samples, the first one is a baseline and the last twenty are useful samples with specific emotions. The PSD feature is obtained by the deviation of PSD from useful samples and the baseline. Afterward, the PSD features of one trial with 32 channels and four frequency bands are processed into the shape of $W_1 \times 128$, where $W_1 = 800$. In addition, we only use the assessment scores of arousal and valence as the labels for each trial.

The SEED IV [40] database records the EEG signals of 15 subjects (7 males and 8 females) through inviting them to watch emotionally stimulating videos. Each subject has to watch 24 stimulating videos with four emotional states (happy, sad, fear, and neutral) for about $n_i$ seconds on three different days. This means that the SEED IV database has three sessions, each with 15 trials. For each participant, the EEG signals are recorded with 62 channels and a 1000 Hz sampling rate, and the labels of the recorded EEG signals are the emotion states of the corresponding video.

In our experiment, we use the preprocessed dataset "de_movingAve" from the SEED IV database. The preprocessing includes downsampling the EEG signals to 200 Hz, removing the noise, eliminating the artifacts, and extracting the differential entropy (DE) feature with a time window of four seconds without overlap. Through the transformation provided by Shi et al. [41], the PSD features can be calculated by:

$$h(X_i) = \frac{1}{2}\log(p(X_i)) + \frac{1}{2}\log(\frac{2\pi e}{n}) \tag{4}$$

where $n$ is the length of the specific time window, $h(X_i)$ is the DE feature, and $p(X_i)$ is the PSD feature. As the preprocessed data have 310 features of 62 channels with 5 frequency bands and the video length is distinguished in each time, the transformed PSD feature has the shape of $W_2 \times 310$, where $W_2$ is the sample numbers of one subject and those in the three trials are 851, 832, and 822, respectively.

*4.2. Experimental Setting*

In this paper, we only consider the subject-dependent pattern. Suppose the preprocessed data of each subject are expressed as $Data = [D_1, D_2, \ldots, D_b] \in R^{N \times (b*m)}$, where $N$ is the trial sample number, $b$ is the frequency band number, $m$ is the channel number, and $D_i \in R^{N \times m}$ is the corresponding dataset of the $i$-th frequency band.

In the DEAP database, the labels of arousal and valence are concerned in our research. Considering that these labels have scores between one and nine, the median of five is set as the threshold to distinguish the scores of low and high labels. Thus, in the binary-classification problem of the DEAP database with $b = 4$, the EEG data of each subject can be expressed as $Data = [D_1, D_2, D_3, D_4]$, and can be divided into 10 sub-datasets. For example, the sub-dataset corresponding to the combination of alpha and beta is $D_{23} = [D_2, D_3]$, and that of beta and gamma is $D_{34} = [D_3, D_4]$. In the SEED IV database, there are four emotional labels. Thus, in the four-classification problem of the SEED IV database with $b = 5$, the EEG data of each subject are expressed as $Data = [D_1, D_2, D_3, D_4, D_5]$, which can be divided into 15 sub-datasets. For example, the sub-dataset corresponding to the combination of theta, alpha, and beta is $D_{234} = [D_2, D_3, D_4]$. Each sub-dataset is used to train the HCRC method in the same way.

Our framework aims to generate combinations of adjacent frequency bands to acquire prediction results for all the combinations through HCRC and fuse the decisions according to these prediction results by CGEL. For example, for one subject in the SEED IV database, the processed EEG dataset is divided into a training set and a testing set, and the training set and testing set are divided into 15 subsets each, which correspond to the 15 combinations. All of the training subsets and testing subsets are then consistently and separately fed into the HCRC method to obtain prediction results, which are noted as column vectors. The training and testing decision sets are then constructed by concatenating all the training and testing subsets by column. The final results of the testing decision set, which are also the results of the testing set, are computed by putting the decision sets into the CGEL method.

In the HCRC method, there is only one regularization parameter λ. In terms of statistics, HCRC does not change the principles of CRC_RLS. The regulation parameter λ produced the best classification outcomes in the range [0.1–1 × $10^{-6}$] when Zhuang et al. [27] assessed the effectiveness of the CRC RLS method. Therefore, λ is set to 0.015 in both datasets. In the CGEL method, we set 100 as the threshold, and, respectively, set (9,2), (8,2), (7,2) and (14,2), (13,2), (12,2) as the window size and stride on DEAP and SEED IV databases. In addition, after the first step in our framework, there are 10 or 15 prediction results; we directly calculate the simple majority voting by row and record it as HCRC-SMV. In this article, we compare the prediction results of HCRC-CGEL with HCRC-SMV, KNN, SVM, and RF in the same training sets and testing sets in our experiments. In the KNN method, the classification results of the DEAP and SEED IV databases are conducted by the function neighbors.kNeighborsClassifier in the Python package sklearn, with the parameter n_neighbors set to 10. In the SVM method, the penalty parameter C in the DEAP and SEED IV databases is, respectively, selected from [0.05, 0.1, 0.5, 1,5] and [0.001, 0.005, 0.01, 0.05, 0.1] by the python sklearn packag's model_select.GridSearchCV function to choose the best. In the RF method, the classification results of the DEAP and SEED IV databases are conducted by the function ensemble.RandomForestClassifier in the Python package sklearn, with the parameter n_estimators set to 100 and random_state set to 1234. In addition, we also compare other neural-networks-based SOTA methods to evaluate the advantages of our framework.

The experiment in this paper involves running Python software on a Mac system using the Core i5 processor.

### 4.3. Statistical Analysis

In this paper, precision, recall rate, and accuracy are used as evaluation indicators of the experimental results of different methods, and the corresponding formula is calculated as follows:

$$\text{Precision} = \frac{\text{TP}}{\text{TP} + \text{FP}}, \tag{5}$$

$$\text{Recall rate} = \frac{\text{TP}}{\text{TP} + \text{FN}}, \tag{6}$$

$$\text{Accuracy} = \frac{\text{TP} + \text{TN}}{\text{TP} + \text{FP} + \text{TN} + \text{FN}}, \tag{7}$$

True positive (TP), true negative (TN), false positive (FP), and false negative (FN) represent the number of positive data points that the framework predicts to be positive, the number of negative data points that the model predicts to be negative, the number of negative data points that the model predicts to be positive, and the number of positive data points that the model predicts to be negative, respectively. In addition, the standard deviation of all the subjects is used to assess the stability of the prediction results.

### 4.4. Performances on DEAP Database

In this section, the five-fold cross-verification prediction results are verified in arousal and valence on the DEAP database through the comparison of HCRC-CGEL with SVM, KNN, RF, and HCRC-SMV. Figures 5 and 6 show the prediction results of arousal and valence on 32 subjects, respectively. As shown in these two figures, the overall trend of each method is roughly the same in arousal and valence. Compared to SVM, KNN, and RF, HCRC-CGEL and HCRC-SMV have the highest prediction accuracy in all the subjects. In addition, the prediction accuracy of HCRC-CGEL in arousal and valence is higher than that of HCRC-SMV on most subjects. However, the prediction results of SVM are unstable. This may be the reason that the SVM does not find the optimal classification surface in the binary classification in some subjects.

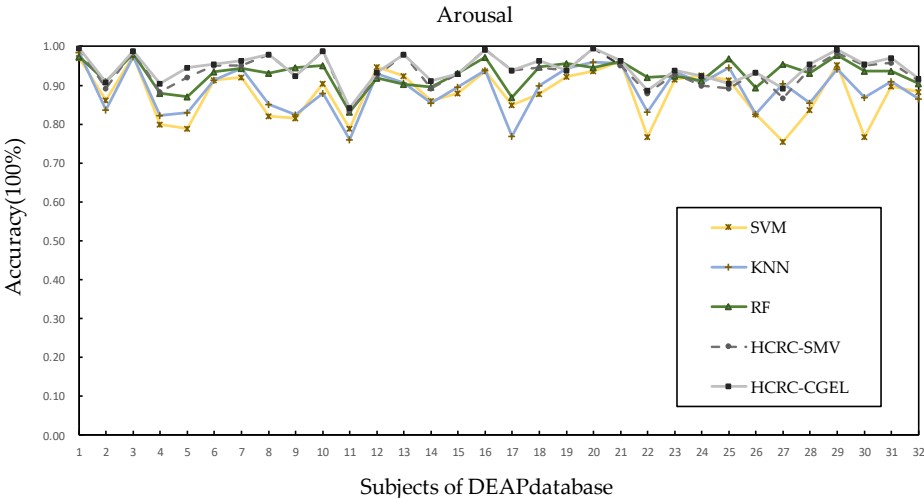

**Figure 5.** Results of each method for arousal on DEAP database.

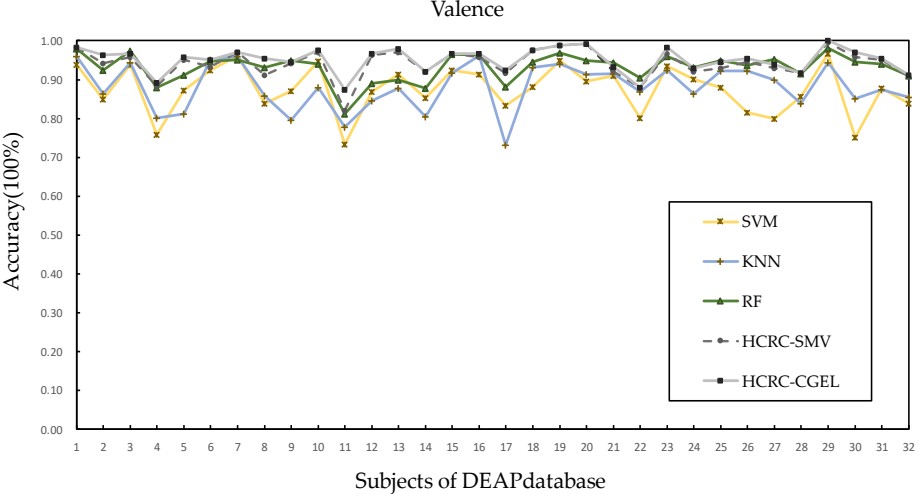

**Figure 6.** Results of each method for valence on DEAP database.

The average accuracies of arousal and valence on all subjects of each method are shown in Table 2. The prediction accuracies of HCRC-CGEL are 94.93% in arousal and 95.09% in valence, which are much higher than those of SVM, KNN, and RF. Compared with HCRC-SMV, the accuracy rates of arousal and valence are increased by 1.05% and 0.83%, respectively. This proves that in the DEAP database, our decision fusion method outperforms simple majority voting. In the prediction results of the compared method, RF performs better than KNN and SVM on almost every subject, and the mean accuracies of RF are higher than those of KNN and SVM on the arousal and valence of the DEAP dataset. Figure 7 shows the confusion matrices of HCRC-CGEL on the DEAP database. The accuracy of low arousal and valence is, respectively, 93.41% and 96.08%, and that of high arousal and valence is, respectively, 94.19% and 95.82%. In addition, the precision and recall rates of arousal are, respectively, 95.06% and 94.75%, and those of valence are, respectively, 94.65% and 95.33%. The standard deviations of HCRC-SMV and HCRC-CGEL are 3.33 and 3.00 on arousal, and 3.40 and 2.90 on valence, respectively, which is lower than those of other methods. This means that our framework has good results on the DEAP database.

As shown in Table 3, our framework is compared with the SOTA methods of other researchers on the DEAP database. For fairness, only the emotion recognition results from the binary classification problem are considered in the comparison group. The results demonstrate that our framework outperforms the SOTA methods in terms of average prediction accuracy in the DEAP database, and that our decision fusion method outperforms

SMV. Consequently, the framework for using the combinations of the adjacent frequency bands can offer greater potential to obtain good results.

**Table 2.** Performance (%) of each method on DEAP database.

| Dataset | RF | KNN | SVM | HCRC-SMV | HCRC-CGEL |
|---------|------|------|------|----------|-----------|
| DEAP(A) | 92.86 | 88.94 | 87.81 | 93.88 | 94.93 |
| DEAP(V) | 93.14 | 88.13 | 87.48 | 94.26 | 95.09 |

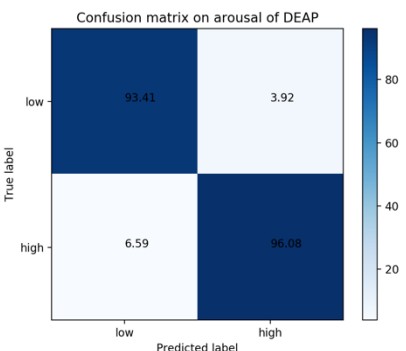 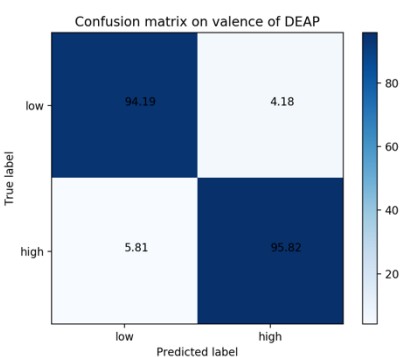

**Figure 7.** The confusion matrix in the arousal and valence of DEAP databases.

**Table 3.** Mean accuracy in arousal and valence of DEAP database.

| Method | Arousal (%) | Valence (%) |
|--------|-------------|-------------|
| 3DCNER (Zheng et al.) [42] | 84.53 | 83.83 |
| ERDL (Yin et al.) [43] | 85.27 | 84.81 |
| ERHGCN (Zheng et al.) [44] | 88.79 | 90.56 |
| SFE-Net (Deng et al.) [45] | 91.94 | 92.49 |
| CR-GCN (Jia et al.) [46] | 93.46 | 94.78 |
| Our Approach (HCRC-SMV) | 93.88 | 94.26 |
| Our Approach (HCRC-CGEL) | 94.93 | 95.09 |

### 4.5. Performances on SEED IV Database

In this section, the five-fold cross-verification prediction results of a four-category classification problem are verified through the comparison of HCRC-CGEL with SVM, KNN, RF, and HCRC-SMV on the SEED IV database. In Figure 8, the overall accuracies of HCRC-CGEL and HCRC-SMV are higher than those of SVM, KNN, and RF, and RF outperforms KNN and SVM. The prediction results of HCRC-CGEL are better than those of HCRC-SMV for most subjects in the three sessions.

Table 4 shows the average prediction accuracy rates of the five methods. The accuracy rates of HCRC-CGEL in the three sessions are 96.36%, 96.97%, and 97.61%, which are better than those of SVM, KNN, RF, and HCRC-SMV. In addition, Figure 9 shows the confusion matrices of HCRC-CGEL on the SEED IV database. In the four classification problems, except for the neutral emotion classification accuracy of session one, which is 94.41%, the neutral emotion in the other session and all emotions in all the sessions are all over 96%. The standard deviations of HCRC-SMV and HCRC-CGEL are 2.18 and 2.01, respectively. This means our framework has good performance on the SEED IV database.

As shown in Table 5, our framework is compared with the SOTA methods of other researchers on the SEED IV database. For the sake of fairness, only the emotion recognition results of the four-classification problem are considered, and the final results are represented by the average prediction accuracy of all subjects in each session and the corresponding standard deviation. The results show that, compared with the SOTA methods, our framework has higher classification accuracy and lower standard deviations on the SEED IV database.

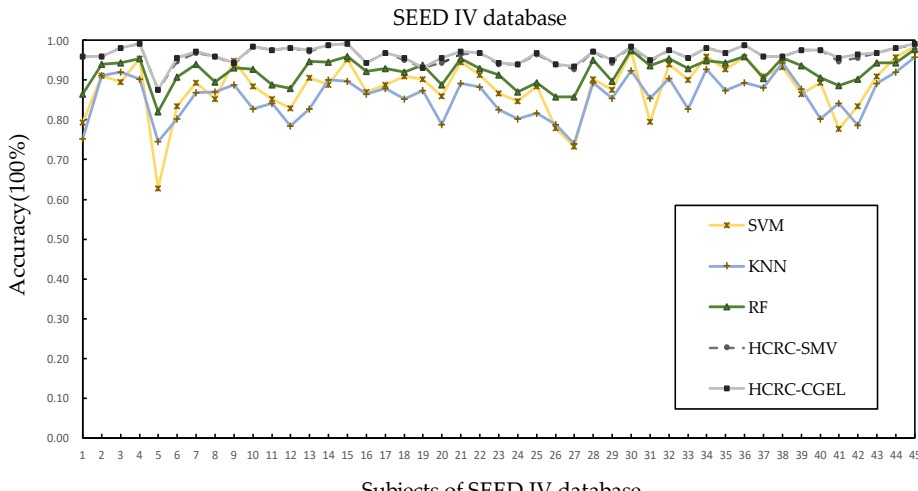

**Figure 8.** Results of each method on SEED IV database.

**Table 4.** Performance (%) of different methods in the three sessions of SEED IV database.

| Dataset | RF | KNN | SVM | HCRC-SMV | HCRC-CGEL |
|---------|-------|-------|-------|----------|-----------|
| SEED IV 1 | 91.64 | 84.94 | 86.84 | 96.25 | 96.36 |
| SEED IV 2 | 91.31 | 84.53 | 87.67 | 96.80 | 96.97 |
| SEED IV 3 | 93.51 | 87.89 | 90.28 | 97.52 | 97.61 |

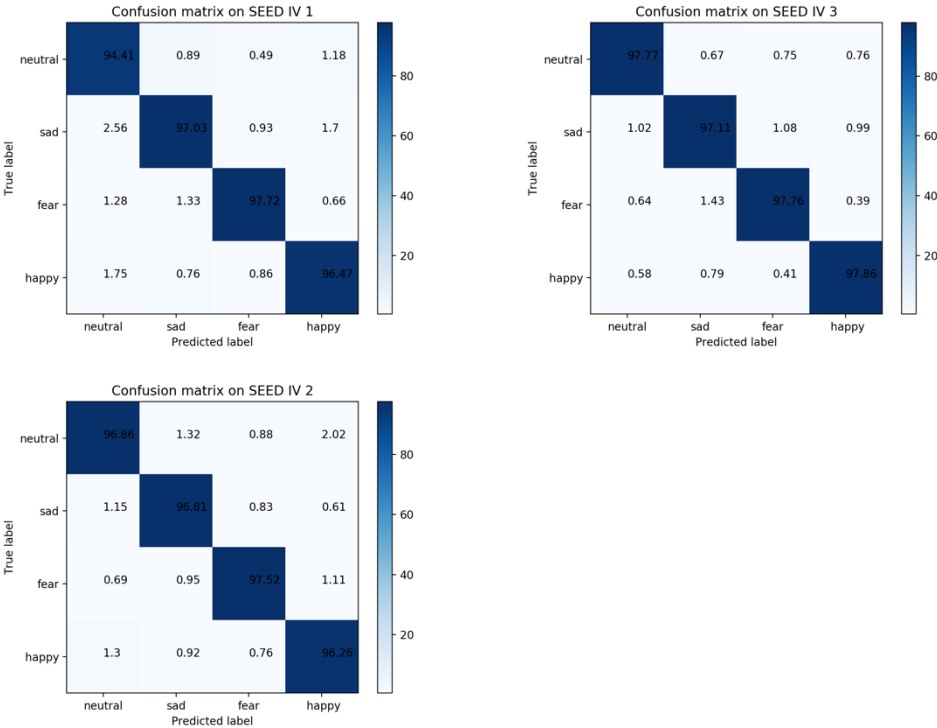

**Figure 9.** The confusion matrix in the three trials of SEED IV databases.

As this section only discusses the decision fusion results of 15 combinations of all adjacent frequency bands, we have published the prediction results of each combination in some used datasets (the arousal of the DEAP database and the third session of the SEED IV database) on Github (https://github.com/zipore/HCRC-CGEL, accessed on 21 August 2022). The 5-fold cross-validation results indicate that the accuracy of the prediction

increases with the number of adjacent frequency bands used. This can also illustrate that the frequency bands have complementary information to each other.

**Table 5.** Mean accuracy and standard deviations (Std) on SEED IV database.

| Method | Accuracy (%) | Std |
|---|---|---|
| MSFBEL (Shen et al.) [30] | 82.97 | 11.06 |
| BDAE (Zheng et al.) [40] | 85.11 | 11.79 |
| DCCA (Qiu et al.) [47] | 87.45 | 9.23 |
| CAN (Qiu et al.) [48] | 87.71 | 9.74 |
| Our Approach (HCRC-SMV) | 96.86 | 2.18 |
| Our Approach (HCRC-CGEL) | 96.98 | 2.01 |

## 5. Discussion

This paper proposed an emotion recognition framework aimed at generating adjacent frequency band combinations to obtain all of their prediction results through the HCRC method and then using the CGEL method to fuse the prediction results. Through the good performances of HCRC and CGEL, the framework could combine complementary information from different combinations to achieve better classification results. This had a significant impact on the accuracy of EEG-based emotion recognition.

The CRC_RLS method was initially proposed to recognize emotions in face data, and encouraging results were obtained in the classification of other types of signals, such as EEG signals and oral odor signals. In terms of statistics, HCRC did not change the principles of CRC_RLS. As a result, our framework could also be applied to those various pattern classification problems. The HCRC method classified the samples of the testing set by taking the value of the representative error of each category of the training set, implying that a large-sample-size dataset was not required. Consequently, in HCRC, we used simple random sampling to balance the samples. Compared with CRC_RLS, the sampling step of HCRC could unify the length of the representation coefficient of each category, which could result in a more accurate classification.

In decision fusion, the results of CGEL were better than SMV in almost every subject. This was benefited by the fact that CGEL used SMV to scan the decision set and ensemble learning to obtain classification results. By learning the characteristics of the training set, CGEL made the classification result of the testing set as close as possible to the self-testing result of the training set. In terms of structure, CGEL adaptively decided the number of layers to train a different structure for each subject, which could improve prediction accuracy and save the calculation costs.

However, there were some limitations. According to the property of collinear vectors, the process that the HCRC method used the regularized residual of the training set to achieve classification did not require many samples. Therefore, HCRC rejected some samples by simple random sampling. While this operation had a positive impact on public databases such as SEED IV and DEAP due to their relatively uniform sample distribution, it might not have the same positive effects on datasets with drastically uneven sample sizes. Moreover, in order to calculate the representation coefficient, a large number of matrix inverse calculations were necessary, and the size of the matrix was proportional to the training sample size. This means that HCRC was only suitable for datasets with small sample sizes. The advantages of CGEL were limited to some degree in the subject-independent pattern. For large sample size data, CGEL might produce superior decision fusion results.

The combinations of all adjacent different frequency bands could make use of the complementary information to a certain extent, but the correlations of frequency bands in different brain regions still needed to be refined. For example, taking the frontal lobe, the parietal lobe, the temporal lobe, and the occipital lobe into account, appropriate channels or brain regions selection for each adjacent frequency band combination could be used to design a more accurate emotion recognition framework. Our experiment was conducted on

the SEED IV database with the tags of sad, happy, neutral, and fear, and the DEAP database with the tags of arousal and valence. In order to demonstrate the applicability of the suggested method, the subject of the dataset could be enlarged to include more examples from different cultural backgrounds, and the tag could be given a fuller emotional response (valance and arousal, positive and negative). Following this, a more comprehensive discussion of emotional space could be proposed.

In this paper, the PSD feature was used because it could represent the distribution and energy strength of signal power over a frequency range [49]. However, the statistical features, Fourier and wavelet-transform-based features, and some deep-learning-based features, all of which were very effective in EEG-based emotion detection models, could also be utilized to support the validity of the proposed method. As the distribution of samples in the experiment's databases was relatively balanced, there was a good prediction result from HCRC in randomly selecting (and rejecting simultaneously) some samples to keep the sample number of each category constant. However, in datasets with a highly imbalanced sample size, this operation might not produce satisfactory results. In the following work, CRC_RLS could be optimized by comparatively selecting a more suitable method to solve the imbalance of sample size. With the limitation of HCRC on sample size, our framework was based on the subject-independent pattern, and this made the advantage of CGEL insufficient. The classifiers that were suitable for the subject-dependent pattern could be considered to achieve more advantageous decision fusion results by the CGEL method. In addition, the specific working mechanism of EEG signals was still not clear, which is also one of the main goals of our future work.

**Author Contributions:** Material preparation, data collection, and analysis were performed by L.Z. and Z.Z. The first draft of the manuscript was written by Z.Z. and all authors commented on previous versions of the manuscript. L.Z. and Z.Z. made critical revisions to the work. All authors have read and agreed to the published version of the manuscript.

**Funding:** This research was funded by the National Natural Science Foundation of China, grant number: 71874126.

**Institutional Review Board Statement:** Not applicable.

**Data Availability Statement:** The preprocessed datasets are available on Github: https://github.com/zipore/HCRC-CGEL, accessed on 21 August 2022.

**Acknowledgments:** We wish to thank all authors for their valuable efforts, and Hongsong Xue (Business School, Wu-han Qingchuan University, No. 9, Yuping Avenue, Longquan Road, Jiangxia District, Wuhan, Hubei, China) for his important contributions, which helped improve the quality of this paper.

**Conflicts of Interest:** The authors declare no conflict of interest.

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
