# Peer review of "A Two-Step Framework to Recognize Emotion Using the Combinations of Adjacent Frequency Bands of EEG"

_applsci, doi:10.3390/app13031954_

Round 1

Reviewer 1 Report

This paper designs a new EEG-based emotion recognition framework, which effectively utilizes the basic information of the frequency band to improve the accuracy of EEG emotion recognition. It is proved that the performance of the scheme is better than the existing technology.

Before considering publication, some issues must be addressed. If the following problems are well solved, the commentator believes that the contribution of this paper is very important for the field of EEG emotion recognition.

1.  Formula symbols are inconsistent. In 171 lines, the symbols in the formula are inconsistent with the previous ones, and a comma is added ;

2.  Image expression is not clear.

1)The represent vectors part in Figure 2 is omitted and the represent errors are not expressed clearly;

2)The part in Figure 4 that links to the best result  lacks ellipsis and the ellipsis between  and  is superfluous;

3)There are many problems of ellipsis offset in the image;    

3.  Table symbol description is not clear. The Statement of  is unclear in table 1

4.  The accuracy of the DT algorithm on the SEED IV database and the accuracy of the SVM algorithm on the DEAP data set are 20 % lower than the proposed algorithm, and the reference value is low.

Reviewer 2 Report

The paper uses EEG data to conduct cognitive research on human behavior patterns. This research has certain novelties, but there are also writing questions in the paper that need to be supplemented and expanded so that readers can clearly understand the innovation of the research.

Several classification method models were proposed in the study, and the author believes that the classification accuracy of this method is very high, but how to verify such a high classification accuracy, the verification of the method needs to be supplemented and described in detail in the paper.

Reviewer 3 Report

In the manuscript “A two-step framework to recognize emotion…” authors used EEG based HCRC-CGEL classification network for emotion recognition. Their framework is set to generate combinations of adjacent frequency bands  to aquire predictions for all HCRC combinations, and fuse these predictions by CGEL. Authors explained their motivation for designing such learning methods, after analyzing different emotion recognition technologies utilizing EEG in the Section Related work. The developed HCRC-CGEL framework was tested on two public EEG data sets, and compared with several other methods. The superior performance of HCRC and CGEL was shown at data sets with small sizes and relatively uniform distribution.

This is the manuscript with potential impacts on the reliability of new emotion recognition methods, based on EEG.  I recommend this manuscript for publication in Applied Sciences.  

I have two comments for authors to consider before the publication. Check if the downsampling the EE signal was at 128 Hz or 200 Hz, since both numbers are present in the 4.1 Subsection. What would be the argument for giving value 0.015 for the regulation parameter Lambda in the HCRC method?

Round 2

Reviewer 1 Report

1. Lines 313-320 format error

2. The picture layout needs to be modified

Reviewer 2 Report

The paper can be accepted in present form.
